# Skin Cancer Microenvironment: What We Can Learn from Skin Aging?

**DOI:** 10.3390/ijms241814043

**Published:** 2023-09-13

**Authors:** Andrea D’Arino, Silvia Caputo, Laura Eibenschutz, Paolo Piemonte, Pierluigi Buccini, Pasquale Frascione, Barbara Bellei

**Affiliations:** 1Oncologic and Preventative Dermatology, San Gallicano Dermatological Institute, Istituto di Ricovero e Cura a Carattere Scientifico IRCCS, 00141 Rome, Italy; 2Laboratory of Cutaneous Physiopathology and Integrated Center of Metabolomics Research, San Gallicano Dermatological Institute, Istituto di Ricovero e Cura a Carattere Scientifico IRCCS, 00141 Rome, Italy

**Keywords:** BCC, skin, SCC, melanoma, skin aging, CAFs

## Abstract

Aging is a natural intrinsic process associated with the loss of fibrous tissue, a slower cell turnover, and a reduction in immune system competence. In the skin, the continuous exposition of environmental factors superimposes extrinsic damage, mainly due to ultraviolet radiation causing photoaging. Although not usually considered a pathogenic event, photoaging affects cutaneous biology, increasing the risk of skin carcinogenesis. At the cellular level, aging is typified by the rise of senescence cells a condition characterized by reduced or absent capacity to proliferate and aberrant hyper-secretory activity. Senescence has a double-edged sword in cancer biology given that senescence prevents the uncontrolled proliferation of damaged cells and favors their clearance by paracrine secretion. Nevertheless, the cumulative insults and the poor clearance of injured cells in the elderly increase cancer incidence. However, there are not conclusive data proving that aged skin represents a permissive milieu for tumor onset. On the other hand, tumor cells are capable of activating resident fibroblasts onto a pro-tumorigenic phenotype resembling those of senescent fibroblasts suggesting that aged fibroblasts might facilitate cancer progression. This review discusses changes that occur during aging that can prime neoplasm or increase the aggressiveness of melanoma and non-melanoma skin cancer.

## 1. Introduction

Skin cancer is one of the most common types of cancer worldwide, with an estimated 3.3 million cases diagnosed each year [1]. Overall, melanoma incidence is ongoing a dramatic rise, rapidly becoming the fifth leading cancer in males and females in the United States [2,3]. At the same time, non-melanoma skin cancer, which comprises basal cell carcinoma (BCC) and squamous cell carcinoma (SCC), are the first and second most common types of skin cancer, respectively [4]. Moreover, these data probably represent an underestimation considering that both BCC and SCC are not typically reported in cancer registries. BCC and SCC, once defined as “epitheliomas” due to their low metastatic potential and most frequently grouped as non-melanoma skin cancers (NMSCs), both originate from epidermal keratinocytes [5]. Conversely, melanoma is a malignant proliferation of pigment-producing cells known as melanocytes. The microenvironment in which skin cancer arises and progresses is complex and dynamic, composed of various cell types, extracellular matrix (ECM) components, and pleiotropic signaling molecules. The intricate interplay between these factors influences tumor initiation, growth, and spreading. Thus, currently, the framework adjacent to the tumor is considered an integral part of the pathologic condition, and, in the oncologic field, disease descriptions have progressively included surrounding and mixed tissue. During its evolution, the tumor gradually shapes host tissue into a favorable milieu promoting cancer dissemination, therapeutic resistance, and immune evasion [6,7]. A particular feature of cancer stroma is the increased number of mesenchymal or fibroblastic cell types, pathologically activated, indicated as cancer-associated fibroblasts (CAFs). CAFs might originate from resident quiescent fibroblasts, mesenchymal stem cells, endothelial cells, pericytes, epithelial cells, adipocytes, and smooth muscle cells as well as from distant sources, such as the mesenchymal bone marrow stem cells [8,9]. CAFs support the proliferation of cancer cells, angiogenesis, and metastasis [10]. Even so, there still is no evidence that via reconstituting the stromal integrity it is possible to hamper cancer cells. Furthermore, there is no proof that parenchymal injury associated with cancer spontaneously reverts following surgical or medical treatments because CAFs are particularly resistant to undergoing apoptosis [10,11]. While extensive research has been focused on understanding the mechanisms underlying skin cancer development and progression, a deeper understanding of the associated microenvironment is still needed. Particularly, little attention has been dedicated to the definition of common traits of cutaneous neoplasms microenvironment. Interestingly, recent studies have highlighted the role of aging in shaping the skin into a pro-tumorigenic milieu [12]. Correspondingly, the incidence of carcinoma positively correlates with age [13] and progeroid syndrome, a heritable autosomal recessive human disorder characterized by the premature onset of numerous age-related diseases, predisposes to cancers [14,15].

### 1.1. Skin Aging Characteristics

Skin aging is characterized by various changes in both epidermal and dermal layers, including alterations in the ECM composition, cellular senescence, and chronic inflammation [10]. All these changes have been linked to an increased susceptibility to skin cancer [16,17,18]. During aging, the gradual loss of tissue functions and reparative capacity leads to the accumulation of cellular damage and dysfunction [19,20]. This process is both the cause and consequence of the progressive increase in the number of senescent cells. These cells are still highly metabolically active secreting factors included in the so-called senescence-associated secretory phenotype (SASP) capable of diffusing the senescent status to surrounding cells [21]. Hence, physiologically, senescence in a tissue represents primarily a barrier against tumors. In a tumor-centric view, prompt discontinuation of proliferation is the most important functional consequence of oncogene-induced cell senescence. Nevi display senescence signature and rarely develop into melanomas [22,23], presumably because aberrant BRAF signaling induces a growth arrest response longer than expected [24,25]. However, the number of nevi represents a major melanoma risk factor [26] as approximately 25% of melanomas appear in pre-existing nevi [27,28], demonstrating that cell senescence is a reversible condition. Longer telomere length, not restricted to melanocytes, correlates with nevi number and size suggesting a higher replicative potential and reduced senescence of these individuals [29]. Even so, the role of telomere shortening, a likelihood factor for skin aging, in the risk of skin cancer (as well as in other neoplastic diseases) is still not clear [30]. The role of the telomere pathway in melanocyte senescence has been also proposed in a study showing that telomerase activity increases steadily from benign nevi to dysplastic nevi to melanoma [31,32]. Contrary to the general belief that cell senescence can trigger cancer, De Vitis et al., reported that telomeres longer than expected are associated with an increased risk of several cancers, including melanoma [33], and another study demonstrated the risk of NMSCs and melanoma is significantly lessened with telomeres shortening [34,35]. This is consistent with recent clinical reports that long telomeres are associated with increased mortality in melanoma patients [36]. Similar to melanoma, a part of SCC derives from pre-existing actinic keratoses (a common skin condition resulting from long-term UV exposure) because of multiple mutations of keratinocytes generating genetic diversification and microenvironmental alterations [36,37,38]. Sun-exposed skin is a field of pre-procancer evolving clones carrying early driver mutations still maintaining physiological functions until subjected to additional mutation and selection. According to the idea that the accumulation of somatic mutations with age occurs before carcinogenesis, acquired DNA sequence variations have been recently reported in most noncancerous human tissues. Commonly, several clones of keratinocytes with a mutation in the *p53* gene are detectable in histologically normal skin [39,40]. Consistently, Gosselin et al., isolated in vitro transformed keratinocytes from post-senescent keratinocytes after prolonged culturing [41]. Oxidative stress in the context of senescence represents a high risk of mutagenicity that might act as an initial event in tumorigenesis. Furthermore, tumors from older individuals tend to accumulate more clonal mutations [42,43]. Overall, these data demonstrate senescent cells accumulate in aged tissues and might, under specific microenvironmental stimulation, become progressively premalignant and malignant [44,45,46]. Therefore, anti-aging strategies are considered suitable for delaying cancer occurrence and improving prognosis [47,48].

### 1.2. Skin Aging Drivers

Aging of the skin is a multifactorial process that involves mechanisms triggered by intrinsic and extrinsic factors that work in concert and influence each other (Figure 1).

Intrinsic aging involves biochemical degenerative processes that progressively take place during life whereas extrinsic aging is a biochemical and physical phenomenon driven by acute or chronic exposition to external stress. Both chronological aging and extrinsic aging are affected by the specific genetic background that includes gender. Of interest, polymorphic variants of the *MC1R* gene, one of the most important pigmentary-related genes, negatively impact both photoaging and skin cancer risk [49,50,51,52]. Protection against cancer is not a simple secondary event related to melanin synthesis and distribution since activation of this membrane receptor by its ligand α-melanocyte-stimulating hormone (α-MSH) exerts pleiotropic effects such as stimulation of intracellular antioxidant machinery, antagonization of inflammation, and modulation of collagen metabolism [53,54]. Other genetic variables involve SNPs of genes related to susceptibility to oxidative stress [55]. There is no doubt that ultraviolet (UV) irradiation exposure, in particular, UVA and UVB represent [55] the major cutaneous stressful factor; however, smoking, pollution, and diet might also play a relevant role [56,57,58]. Interestingly, while the effect of most of the environmental factors (environmental exposome) largely depends on lifestyle, the side effect of ultraviolet exposition mostly depends on skin phenotype [59,60,61]. UVA and UVB induce different whilst overlapping biological responses including abnormal reactive oxygen species (ROS) accumulation and/or DNA damage in both the epidermal and dermal compartments of the skin [62]. Hence, in habitual sun-exposed areas, photoaging overlaps intrinsic aging resulting in cumulative detrimental effects. Photoaging accounts for approximately 80% of facial aging [63]. Beyond cancer, aged skin is susceptible to microbial infections, autoimmune disorders, senile pigmentary changes, and benign proliferation such as seborrheic keratosis and lentigo maligna [64].

Oxidative stress resulting from both elevated ROS production and a decline in the antioxidant network is a hallmark of aging biology [65]. The production of ROS during natural aging is moderate, mostly linked to the quantity and the quality of mitochondria activity, and the related signaling pathways are activated slowly and moderately [66]. Under extrinsic pro-oxidant stimulation, relevant damage-associated signaling pathways (DAMPs) are mobilized promptly and fiercely, a fact that might affect cutaneous architecture and function. Hence, in the skin, UV radiation, the main etiological factor for skin cancer [67], impacts both pre-cancerogenic and resident stromal cells complicating the scenario (Figure 2). Thus, differently from internal organs, in the skin stroma alteration can precede (or act independently) tumor onset moving as a driver of the tumorigenic process.

### 1.3. Senescent Fibroblast and Cancer-Associated Fibroblast Overlapping Features

According to the idea that tumor-stroma cross-talk in the skin might not be the only factor responsible for stromal cell reprogramming, sun exposure might directly cause changes in dermal fibroblasts resulting in increased growth factors secretion and the acquisition of the senescent-like phenotype resembling, at least for some aspects, that observed in CAFs [68]. Studies comparing skin-aging-associated secreted proteins (SAASP), and the more general SASP biological processes demonstrated some peculiarities and a partial overlap mostly related to the enriched expression of chemokines and interleukin-signal transducers, IGF activity regulators (IGFBPs), and matrix metalloproteinases (MMPs) [69]. Thus, it is possible that dermal fibroblast of chronically sun-exposed areas could be primed for tumor cell-dependent activation. However, the association of CAF phenotype to those of senescent fibroblast might be ambiguous since, many patient-based studies have documented how a high CAF number is linked to an increased rate of local recurrence and an overall worse outcome in several tumors [70], whereas, it is generally accepted that there is a decrease in the number and proliferation of fibroblasts in the skin with age [71,72]. A possible explanation might reside in CAF’s resistance to apoptosis that differently to normal fibroblasts tends to accumulate over time [73]. Interestingly, Sharathchandra and collaborators demonstrated that the expression and activity of p53 are decreased in CAFs. The loss of p53 allows the fibroblasts to overcome senescence and promotes stromal and cancer cell expansion [74]. Accordingly, another study discovered that the conversion of normal fibroblasts into CAF involved the alteration of non-mutated p53 [75,76]. One more research group reported a coordinated expression of p53 and its target gene p21 in cancer cells and adjacent CAFs within tumor tissues as well as comparable modulation of this important oncogene under DNA damaging compound confirming the control of tumor cells on their immediate microenvironment [77].

This review presents an overview of common skin features acquired during age presenting a gain advantage in the early process of skin oncogenesis. Moreover, differences and similarities between senescent fibroblast and CAFs will be discussed to explore potential therapeutic opportunities that may arise from our understanding of skin aging in the context of the skin cancer microenvironment.

## 2. Materials and Methods

This manuscript combined data from reviews and experimental articles with the thematic of skin aging and skin cancer in humans with a specific emphasis on articles comprising both topics. The analysis covered a wide recent and current literature without any restriction related to the publication time. As this review has been designed as a narrative review, it lacks systematic methods. Considering the enormous material published in skin aging and skin cancer fields, to avoid the limitation of selection bias the author’s list included researchers with different expertise.

## 3. Skin Aging Phenotype

Natural aging impacts the human body differently depending on the highly complex organization of organs, tissue, and cells. The human skin is the largest body organ offering the first line of defense against environmental factors and modifications of the skin as well as of associated appendages represent the most visible sign of aging. Consequently, the standardized phenotypic representation of aged phenotype is frequently referred to as features easily visible such as irregular pigmentation, wrinkling, telangiectasia, hair graying, hair loss, diminished elasticity, tonus loss, and dryness caused by structural changes that do not necessarily find obvious correspondence to the important functional tissue reshape involved in skin cancer biology. By contrast, less evident at the macroscopic level, molecular aberrations related to key intracellular pathways such as mitochondria metabolism, cell membrane defects, reduced repair processes, and skin defective signaling to immune system cells consistently influence disease susceptibility including tumors. Here, we will focus on functional aspects of skin aging discussing their possible involvement in oncogenesis.

### 3.1. Age-Related Modification of Skin Paracrine Network

The secretome of human cells is comprised primarily of cytokines, growth factors, hormones, enzymes, glycoproteins, lipid mediators, and membrane-encapsulated extracellular vesicles (exosomes and micro-vesicles). Extracellular vesicles include mainly RNA species (microRNAs, long non-coding RNAs, mRNAs), lipids, DNA fragments, and other cellular debris [78]. Due to their high stability, exosomes are present in all body fluids and deliver the included material in autocrine, paracrine as well as endocrine manners. Thus, by loading specific SASP soluble components onto exosomes, senescent cells can spread senescence to proximal and distant cells [79]. SASP reinforces cell cycle arrest, drives the immune-mediated clearance of senescent cells, triggers the epithelial-mesenchymal transition, and influences tissue-remodeling activities [80]. Senescence repercussions differ depending on the cell-type specific proliferation capacity. Rapidly dividing cells, such as epidermal keratinocytes are continuously replaced due to normal turnover, whereas fibroblasts, having a limited proliferation rate, in response to stress accumulate functional defects and undertake senescence programming impacting tissue integrity [71,72]. Additionally, senescent keratinocytes are constantly removed by macroautophagy, a process that if not completed might facilitate the evolution of transformed cells [41,81]. Melanocytes are intermittent mitotic cells, which express the senescence markers such as p16, HMGB1, and telomere shorting during aging [82]. Further, great importance has been attributed to senescent melanocyte SASP due to its capacity to induce telomere dysfunction in surrounding cells and to impair basal keratinocyte proliferation [82]. However, senescence phenotype in the melanocyte lineage depends on several variables including the phototype, the body district (sun-exposed or sun-unexposed) and the life style. Thus, it is considered that a valid indicator system for human aging studies is long-living post-mitotic fibroblasts, the most abundant cell type of the dermis. Chronically driven aging is a slow and steady accumulation of defects frequently associated with the compensative reorganization of intracellular functions, whereas stress-induced premature senescence is induced by acute events requiring rapid adaptation to a mutated condition. Dermal fibroblasts are widely analyzed in vitro for senescent signature characterization, including the SASP profile. However, it is important to take into consideration that the description of naturally aged fibroblasts does not fully overlap with the stress-induced premature cell senescence phenotype [12]. In fact, in apparent contradiction to the association between SASP and aging, dermal fibroblasts with age tend to reduce their secretory activity [83]. Over the course of life, fibroblasts become flattened cells with a dense endoplasmic reticulum, voluminous fascicles, or bundles of microfilaments into enlarged cytoplasm and become quiescent cells [84]. Moreover, lower fibroblast cellularity in the elderly, reduction in proteostasis, lower receptor numbers, and responsiveness argue for an attenuated fibroblast communication network [83] suggesting that naturally aged dermis could not be intrinsically pro-tumorigenic in the absence of tumor cell-derived reprogramming stimuli. On the other hand, attenuated secretory activity by old fibroblasts may lessen the number of regulatory factors physiologically produced by mesenchymal cells in the skin compromising their natural anti-tumoral protective role. This latter mechanism might play an important role in early tumor onset, whereas tumor-driven reprogramming of stromal cells might be relevant in the second phase of cancer progression. Also, sebaceous gland secretion decreases with age even if the number of sebaceous glands remains unmodified [85]. This is partially explained by the dissociation of sebaceous gland cell proliferation and differentiation observed in aged skin [86]. With aging, enlarged sebaceous glands size and gender differences evidence strong hormone dependency [87,88]. The levels of skin surface lipids in the elderly return like those of prepubertal age, and generally the mean sebum levels in females are diminished more than in males [86]. The damaging effect of UV on the sebaceous has been attributed to UVA; UVA penetrates deeper into the dermis and can reach the sebaceous gland [89]. Aging of sebaceous glands, especially in sun-exposed areas, starts with compensative hyperplasia followed by atrophy which could in some cases develop into sebaceous gland carcinoma [85,90]. As an integral part of the skin inflammatory process, sebaceous glands regulate innate immune response directly releasing various inflammatory mediators and indirectly contributing to the lipid milieu composition of the dermis, which impact the immune phenotype of various cell [6]. Thus, under stimulation sebocyte-derived proteins and lipids may modify the skin microenvironment facilitating skin cancer onset. With age, skin barrier functionality is compromised reducing the ability to manage external injury, to resolve chronic inflammation predisposing to uncontrolled proliferation. At the same time, UVB stimulates the secretion of IL8 and IL1β in human sebocytes in vitro [91]. Aging affects the production of proopiomelanocortin (POMC) and the processing of derived peptides which are key players in skin biology. Histologically, geriatric skin shows a reduction in the number of nerve endings and microvessels [92]. The peripheral nervous system locally releases neuroendocrine factors (catecholamines, adrenocorticotropin, substance P, somatostatin, as well as neurohormones, such as proopiomelanocortin-derived peptides, α-MSH) having implications in tumor initiation and progression [93]. UV and other stressful conditions stimulate sensory nerves to release nerve growth factors (NGF) and substance P [94]. Interestingly, substance P has been associated with the inflammatory cascade and in NMSC due to the transactivation of epidermal growth factor receptor (EGFR) [95]. Confirming the idea that a firm brain-skin connection, exists (as suggested by altered skin physiology in patients with neurodegenerative diseases), lower MCRs on cell membranes of aged brains correlate with the age-related decline of cognitive function [96].

#### 3.1.1. Growth Factors

In extensive research, Waldera-Lupa and collaborators, using an in situ proteomic approach, identified a set of proteins with a specific age-associated abundance change related to specific biological processes: proteostasis, proliferation, cell differentiation, cell death, cytoskeleton organization, response to stress, cell communication, and signal transduction, as well as ribonucleic acid metabolism [69]. The same group, in an in vitro study, delineated the age-dependent secretion patterns indicated as ‘‘skin-aging-associated secreted proteins”, or SAASP, of dermal fibroblasts [97]. These proteins, along with the molecules involved in elastic fiber formation, glycosphingolipid, and sphingolipid metabolism, partially differ from the general SASP characterized in vitro. Interestingly, growth factors, which are critical for cancer progression [98,99] are not included in these lists [69,97]. In line with these data, growth factors are not comprised in a more general frailty biomarkers list regulated during aging or age-related diseases [100,101]. The unique exception is represented by growth differentiation factor 15 (GDF15) [101,102], a member of the transforming growth factor-beta superfamily, implicated in mitochondrial dysfunction, cancer growth, and epithelial-to-mesenchymal transition [103]. Being a potent marker of cell stress, GDF15 is a biomarker of SASP, and knocking down its expression in dermal fibroblast promotes premature senescence [104]. Augmented GDF15 expression by UV radiation in fibroblast is implicated in aging-associated hyperpigmentation [105]. P53-dependent transcription of the *GDF15* gene is also part of the UVB response of melanocytes [106]. Moreover, a low level of GDF15 in the epidermis of psoriatic patients correlates to accentuated inflammatory signals in the skin [107]. Thus, GDF15 is an essential element of cellular stress response whose lack facilitates the acquisition of the senescent phenotype. As a target of the MAPK pathway, GDF15 is overexpressed in benign nevi, primary and metastatic melanoma [108]. However, high levels of circulating GDF15 correlate to depth of tumor invasion and metastasis and poor patient outcome [108,109]. Furthermore, RNA-seq analysis of ionizing radiation-induced senescent dermal and epidermal cells did not induce modulation of growth factors [45]. By contrast, SAASP and SASP share an enriched expression of IGF activity regulators, particularly IGF-binding proteins (IGFBPs) [97]. During the aging process, the dermis significantly reduces the production of IGFs [110]. Using an organotypic in vitro model Mainzer and collaborators, demonstrated that lessening of IGF1 production reduces the adhesion, proliferation, and capacity to counteract oxidative stress of keratinocytes [111]. Age-dependent changes in IGF activity are a general state not restricted to the skin. A decline in IGF1 serum levels in older adults associated with an augmented age-associated disease risk (diabetes, dementia, vascular diseases, and osteoporosis) further suggests that IGF1 might be a modulator of a common underlying pathway for several age-associated conditions and, potentially, aging itself. Differently, this effect was not evidenced for cancer risk probably due to the IGF effect on intracellular pathways as the IRS/Akt/MAPK axis is critically involved in cell proliferation [112]. IGFBPs exert two opposite effects on IGF bioavailability: as the transporter increases IGF half-life and distant distribution, high-affinity binding partners regulate IGF access to their receptors [113,114]. The secretion of IGFBP7 is considered a dominant mechanism in inducing senescence in the melanocyte lineage [115]. Treatment with recombinant exogenous IGFBP7 human melanoma cell lines carrying mutated *BRAF* can induce cell senescence and apoptosis [115,116], and systemically administrated in xenograft mouse models counteract melanoma growth [115,117]. Interestingly, the expression of IGFBPs is higher in melanocytes [118] and fibroblasts [119] of Vitiligo, a skin autoimmune condition presenting several markers of age-independent skin premature senescence [118]. IGF1/IGF1R signaling is protective against photocarcinogenesis [120,121]. Activation of IGF1R in keratinocytes favors a senescent state which is useful for DNA damage repair and preventing the accumulation of mutations in UV-stressed cells. Inhibition of IGF1R by ligand withdrawal, treatment with IGF-1R-specific small molecule inhibitors, or neutralizing antibodies before irradiation increased the sensitivity of keratinocytes to UVB-induced apoptosis [113,122]. Keratinocytes that survive if they do not become senescent and do not rescue DNA integrity can continue to proliferate with the potential of initiating malignant transformation [122]. Hence, in geriatric skin diminished IGF1 production by senescent fibroblasts supports carcinogenesis in sun-exposed skin [120,123]. Corroborating a protective role for IGFR engagement against NMSC in keratinocytes, an epidemiological study demonstrated that type 2 diabetic patients under chronic insulin therapy (capable of binding IGFR in addition to its specific receptor) to treat their disease, have a significant age-dependent lower risk of developing NMSC compared to type 2 diabetic patients using non-insulin medications [124]. Like IGFs, mitogens in the fibroblast growth factors group (FGFs) are barely produced by aged fibroblasts a fact that compromises the synthesis and turnover of collagen and elastin [125,126]. The paucity of signaling originating from this class of growth factors is also responsible for the scarce tissue repair capacity typic of the elderly [127]. Besides, age significantly reduces EGF and FGF2 release [128]. It is interesting to note that a study based on dermal fibroblast cultures of donors <65 years did not find significant age-related differences in bFGF, hepatocyte growth factor HGF, and stem cell factor (SCF) expression suggesting that relevant variations occur late during life [129]

Since the hypersecretion of mitogenic molecules is a peculiar feature of CAFs [130,131,132], the up modulation of growth factors appears to be the turning point for pro-tumorigenic switching of normal fibroblasts (Figure 3).

Exposure to environmental factors might represent an important step in this conversion. A single exposure to UVA and UVB can activate bFGF, HGF, and TGFβ1 in fibroblasts [133]. Similarly, in vitro, the photo-induced senescence-like phenotype of fibroblasts achieved by serial UV exposure leads to the increased production of SCF and HGF [134,135]. Endothelin-1 (ET-1) mRNA in keratinocytes and HGF mRNA in fibroblasts are upregulated in sun-exposed skin [129]. In vivo, data demonstrated up-modulation of the same peptides in addition to SCF, KGF, ET-1, and α-MSH in senescent keratinocytes of lentigo senilis (LS), a representative common skin condition of aging characterized by spots with accentuated epidermal pigmentation [136,137,138]. Even if it is probably related to the physiological pro-melanogenic effect exerted by these factors that tend to implement the protection against UV, it is important to consider their well-documented involvement in melanoma which also produces these molecules for the autocrine purpose [139]. C-Met, the HGF receptor, is over-expressed in BCC, SCC, and melanoma indicating the compelling dependence of skin tumors on HGF mitogenic activity [140]. HGF is crucial for melanoma proliferation, survival, motility, invasiveness, distant metastatic niche formation, and the acquisition of resistance to oncoprotein-targeted drugs [139,141]. By contrast, the role of c-kit/SCF signaling in melanoma is still debated since this pathway is more active in benign nevi compared to metastatic melanoma favoring the maintenance of a fully differentiated state of melanoma cells [136,137,138,142,143]. It is well documented that fibroblast conversion into CAF mainly occurs through paracrine signaling with cancer cells given that deregulation of the secretome in oncology comprises the bidirectional tumor-stroma cross-talk. The intense release of bioactive molecules by pathological tissue explains not only the modification of fibroblasts within or directly in contact with the tumor but also of those from cancer-free tissue adjacent to the tumor [144]. The reciprocal exchange of messengers also includes mitogens. For example, melanoma synthesizes FGFs, that due to the frequent constitutive activation of the downstream mitogen-activated protein kinase-extracellular signal-regulated kinase (MAPK-ERK) signaling pathway, do not confer a specific advantage to tumor cells but can influence the neighboring cells in the tumor niche, such as endothelial cells and fibroblasts contributing to intratumoral angiogenesis and the development of resistance to therapeutics [145]. Chronic UV exposure causes TGFβ1/Smad3 signaling activation which contributes to cancer progression by regulating many physiological processes, including cancer cell proliferation, and invasion [146]. Overexpression of TGFβ is one of the most prominent characteristics shared by photoaged skin and skin tumor microenvironments [147]. Of note, TGFβ signaling is a growth inhibitor for normal keratinocytes but UV-induced mutation frequently unable BCC, SCC, and melanoma cells to escape from TGFβ-induced growth inhibition [146]. In this case, deregulated TGFβ has been attributed to p53 mutation, which abrogates the p53-mediated transcriptional inhibition of TGFβ [148]. TGFβ-activated CAFs intensify the secretion of growth factors, including TGFβ reinforcing the associated molecular signaling, FGF2/7, vascular endothelial growth factor (VEGF), platelet-derived growth factors (PDGF), and HGF to promote cancer cell proliferation [147,149]. TGFβ1 signaling deregulation is involved in photo-inflammation, a remodeling of the immune system activity in response to excessive exposure of the skin to UV. Furthermore, TGFβ might act as a suppressor of T-cell-mediated immune surveillance acting as a key molecule for the acquisition of resistance to checkpoint inhibitors [150].

#### 3.1.2. The Immune Microenvironment

Immunity is an integral part of the skin, and a vast array of immune cells is housed within it, such as Langerhans cells (LCs), dendritic cells (DCs), macrophages, monocytes, and T-cells [151]. In addition to infiltrating immune cells, the skin immune system comprises several molecular mechanisms that constitute a sophisticated mechanism that provides physical, chemical, and microbiological barriers, to protect the host from external insults and microbial invasion [152,153]. The aging process has been linked to a reduction in the effectiveness of cutaneous immune surveillance, increasing the susceptibility to skin infections and cancers [154].

SASPs include a wide range of cytokines (IL1α, IL1β, IL6, and IL8), and chemokines (CCL2, CCL3. CXCL1, CXCL5, CXCL8, CXCL10) that cooperate to establish a pro-inflammatory environment and to recruit immune cells into the senescent tissue to remove damaged and transformed cells [147,149,155]. However, the mechanism’s efficacy degrades over time as evidenced by the gradual accumulation of impaired cells, nonfunctional organelles, and other waste products derived from the basal metabolic process (self-debris). Toll-like receptor family (TLR), intracellular NOD-like receptors (NLRs), and cytosolic DNA sensors are triggered by ROS and undesired excess of metabolites (e.g., ATP, fatty acids, ceramides, cardiolipin, amyloid fibers, succinate, oxidized lipids, advanced glycation end-products, mitochondrial matrix portions, and mitochondria DNA) leading to inflammasome assembly, continuous secretion of several proinflammatory mediators and innate immune system recruitment [156,157]. When defects originate from the decline of physiological function, such as during aging, the adaptive mechanism converts the perturbation into a stable rearranged tissue homeostasis. This chronic non-resolutive flawed innate immune response is referred to as “inflammaging” [158]. Inflammaging, defined as sterile low-grade inflammation is part of the natural process of body aging and cannot be completely separated from functional aging of the immune system which is called ‘immunosenescence’. Immunosenescence is a complex process involving multiple reorganizational and developmentally regulated changes, rather than a simple unidirectional decline of complete immune function [159]. On the other hand, some immunological parameters are commonly reduced in the elderly, and reciprocally good function is tightly correlated to health status. Whereas innate immunity is relatively well preserved in the elderly, acquired immunity is more susceptible due to the functional decline associated with time, and to the cumulative antigen burden to which an individual has been exposed during their lifetime [160].

In the skin, immunosenescence is characterized by an increase in neutrophils and mast cells in the dermis and by a lower number of epidermal LCs with decreased migration toward lymph nodes compared to younger controls [161,162]. Moreover, aged skin shows a higher CD4^+^/CD8^+^ T cells ratio which is supposed to predispose to inflammation [163] Nevertheless, immunosenescence correlates to insufficient protective function and the opposite condition of increased autoreactivity of the intrinsically aged immune system [160]. In older subjects, circulating antigen-specific T cells are normal but do not efficiently migrate into the skin [160]. It is evident that in this scenario, inflammaging could be sustained by natural immunosenescence (and it is part of this) as well as by extrinsic factors such as UV. Nevertheless, immunosenescence correlates to insufficient protective function and the opposite condition of expanded autoreactivity of the intrinsically aged immune system. Several studies have shown that senescence-induced inflammation is associated with an elevated number of immunosuppressive T regulatory cells (Tregs) in human skin [164] and a reduction in antigen-specific T cells [164,165]. Moreover, inflammatory changes appear to stimulate counteracting immunosuppression to protect the skin from excessive inflammatory injuries. In line with the idea that the resolution of acute inflammation triggers a prolonged post-resolution immunosuppressive phase associated with repair processes of the inflamed tissue [166,167], it has been demonstrated that UVB exposure induces expression and secretion of IL10 and TGFβ cytokines in keratinocytes [91,168] as well as the abundant release of IL10 by macrophages and neutrophils into human skin [169,170]. However, in the case of hyperproliferative diseases, these mechanisms might contribute to immune surveillance escape. Treg cells characterized by the expression of CD4, FOXP3, and CD25 have a negative prognostic significate in oncology since these cells abrogate T-cell immunity to tumor-associated antigens [171]. The negative effect of Treg accumulation on the induction of effective immune responses in the skin and increased numbers of Tregs have been observed in old skin, primary melanoma, melanoma metastasis, BCC, and SCC of the skin [160,171,172,173,174]. Treg infiltration correlates with tumor invasion deep in BCC [175]. Furthermore, Treg depletion induces tumor rejection [176,177] indicating that modulation of Tregs might be a promising approach in cancer therapy.

#### 3.1.3. Metabolic Intercellular Cross-Talk in Skin Aging

Metabolic alterations in tumors orchestrate tumor immunology by affecting the functions of surrounding immune cells, thereby interfering with their antitumor capacity, in addition to the direct influence on tumor cell-intrinsic biological activities. In nature, several small molecules by-products of basal metabolic activities function as signaling molecules. Among these, ROS plays an important role in several biological processes [178]. High concentrations of ROS are pathogenic and can cause severe damage to organelles, DNA, and proteins whereas moderate amounts of ROS are essential for the maintenance of numerous physiological functions, including gene expression. During natural aging, impaired function of mitochondria represents the principal source of ROS. ROS target mitochondria themself, particularly mtDNA triggering a vicious circle that compromises energy production. The decline in mitochondrial capacity with aging is linked to decreased mitochondrial abundance [179], protein expression [180], and mtDNA [181,182,183,184]. An extensive review of changes in metabolite levels in different aging organs and cell models concluded that a common enhancement of the glycolytic pathway and impairment of the tricarboxylic acid cycle pathway causes a shift of the energy centers from the mitochondria to the cytoplasm reflected by changes in the levels of several metabolites, such as decreased fumaric acid and malic acid accompanied by a relative increase in the level of pyruvate [185]. Regarding the skin, authors concluded that data converged in demonstrating glutamate, glutamine, lactate and isoleucine differences [185]. Further, a general intracellular accumulation of amino acids in the aging phase might be a consequence of decreased protein biosynthesis associated with reduced proliferation in old skin [186]. Prolonged exposure to UV radiation leads to the development of nuclear and mtDNA damage accelerating the gradual intrinsic decline in mitochondrial activity [185,187,188]. Specifically, an important indicator of photoaging, which may be less apparent in chronological aging, is the occurrence of significant deletions in mtDNA [189,190]. Common mtDNA deletions including a 4977 bp and a 3895 bp deletion have been reported in sun-exposed human skin [191,192], and NMSCs [193]. However, additional studies demonstrated that mtDNA deletions and tandem duplications in precancerous actinic keratosis or cancerous skin tissues occur with a frequency like those of chronically sun-exposed normal skin, suggesting that it is a feature of photoaging rather than of photocarcinogenesis [190,191]. These common mitochondrial genome deletions, which remove the region encoding *ATPase8*, *ATPase6*, *COXIII*, *ND3*, *ND4*, *ND4L,* and *ND5* genes, 12 s, and 16 s rRNAs and tRNAs, result in mitochondrial dysfunction [194]. While these deletions have been adduced as predictive markers for photoaging and skin cancer [191,193], their role in the induction of altered metabolism and SCC progression is still not clear. Moreover, the study of habitually sun-exposed skin revealed a 260 bp mtDNA tandem duplication in the regulatory region for both replication and transcription sites of mitochondrial DNA (D-loop) [195]. The instability of the D-loop and point mutation of this region has been reported with different frequencies in primary and metastatic melanoma [196,197]. This type of damage is unexpectedly more frequent in dermal fibroblasts [191]. The observation that the dermis harbors a higher level of mitochondria genome damage than the epidermis is explained by the lower turnover of dermal cells that tend to accumulate damage more than rapidly replaced epidermal cells. The involvement of the cellular response to DNA damage could impair metabolic organ functions and induce local inflammation that disturbs the homeostasis of tissue and systemic glucose metabolism. Several studies conducted by different research groups have demonstrated that augmented glycolysis can result in the indirect activation of the NF-κB pathway in response to DNA damage through post-translational O-GlcNAcylation modification of proteins and the production of inflammatory cytokines [196,197,198]. Hence, the regulation of the NF-κB pathway and glycolysis is bidirectional since the NF-κB-dependent inflammatory profile may stimulate glucose uptake and glycolytic activity in dermal fibroblasts [199]. O-GlcNAcylation is an emerging potential mechanism for cancer cells to induce the proliferation and progression of tumor cells and resistance to chemotherapy [200]. O-GlcNAcylation acts as a key orchestrator of both intrinsic and extrinsic pathways involved in immunosuppression in the tumor microenvironment, thus favoring tumorigenesis [201]. Additional data support the role of O-GlcNAcylation in epithelial-mesenchymal transition, a key step for tumor evolution [202]. Interestingly, Gross and collaborators reported a complicated mechanism that links mevalonate pathway activation by UV irradiation to increased glucose uptake, which is shunted to the hexosamine pathway to increase the O-GlcNAcylation of proteins to promote the malignant transformation of melanocytes and increase the invasiveness of melanoma [203]. A study by Noordam et al. showed that among non-diabetic subjects, higher circulating glucose levels are associated with a higher perceived age independent of sex and insulin level [204]. A possible explanation is that serum glucose levels may cause the induction of premature cellular senescence. Several studies have shown that culturing human fibroblasts under hyperglycemic conditions results in both increased oxidative stress as well as an accelerated appearance of premature cellular senescence [205,206]. Senescent cells exhibit higher glycolytic activity and lactate production than younger cells, by an enhanced expression of lactate dehydrogenase A as well as increases in tricarboxylic acid cycle activity and mitochondrial respiration. The latter is likely due to the reduced expression of pyruvate dehydrogenase kinases (PDHKs) in senescent cells, which may lead to increased activity of the pyruvate dehydrogenase complex [207]. Exacerbated glycolysis has been observed in the photoaged epidermis to be associated with a less proliferating and more differentiating phenotype of the keratinocytes [161]. Accordingly, in vitro, data confirmed the intensification of glucose uptake in old keratinocytes; however, it is mostly converted to lactate potentially to compensate for energy deficits due to defects in mitochondrial respiration [186]. In the human epidermis, augmented glucose levels argues for different level of glucose uptake [208]. On the other hand, dependency on elevated levels of glucose accessibility is typical of cancer cells that are subject to the Warburg effect, by which cells metabolize elevated amounts of glucose through a high glycolytic rate at the expense of increased lactate production and lower oxidative phosphorylation [209,210]. Excess glucose is a leading factor of senescence induction [211,212,213,214]. Notably a sustained hyperglycemic state, such as that occurring during diabetes and obesity has been linked to a higher risk of a variety of cancers including melanoma and non-melanoma skin cancer [215,216], and the therapeutic use of Metformin, a glucose-lowering agent, has been demonstrated to be protective for melanoma risk [217] and patient outcome [218]. Glucose availability drives melanoma cell proliferation through the transcriptional regulation of MITF [219]. Activating mutations in N-RAS and BRAF proto-oncogenes in melanoma increase the glucose uptake by upregulation of glucose transportase-1 (GLUT1) and GLUT3 and enhanced glycolytic rates, as well as modifications of glutamine metabolism [220,221]. Thus, treatment with BRAF inhibitors implies a major dependency of pathological cells on mitochondrial activity [222]. Consequently, metabolic targeting synergizes with MAPK inhibition enhancing the anti-tumor effect [223].

Advanced glycation end (AGE) products are stable compounds generated through a nonenzymatic process known as glycation. This process involves the covalent binding of sugar molecules, such as glucose or fructose, to proteins, lipids, or nucleic acids, resulting in the impairment of their normal function. Approximately 20 different types of AGEs have been detected in human skin [224]. Glycation plays a role in both intrinsic and extrinsic aging processes. Oxidative stress is an important factor contributing to the formation of endogenous AGEs whereas exogenous AGEs are food-, smoke and pollution-derived. AGEs interacting with and their receptors (RAGEs) can directly exacerbate oxidative stress in dermal fibroblast by decreasing the activity of superoxide dismutase [225]. In the epidermis, AGEs reduce lipid content influencing sebum composition and skin barrier function [226] and promoting the production of melanin in melanocytes through the ERK-CREB-MITF axis. AGEs are inflammatory mediators with the ability to induce epigenetic changes in pre-malignant lesions and to silence tumor suppressor genes [227]. AGEs and their receptors are significantly overexpressed in melanoma and associated peritumoral stroma compared to normal skin [227]. Glucose-derived AGEs and glyceraldehyde-derived AGEs sustain melanoma cell proliferation and enhance melanoma cell migration and invasion at the same time [227]. Moreover, anti-RAGE antibody represses the growth of implanted melanoma cells in immunocompetent mice, and treatment with an anti-RAGE antibody ameliorates survival rates in tumor-bearing mice, suppressing spontaneous pulmonary metastases of melanoma [227]. Lessened ATP production capacity occurring during aging represents a fragility marker since enhanced ATP production is part of the adaptation to stress conditions. In response to energy deficiency, AMP-activated protein kinase (AMPK) activates p53 by phosphorylating it at Ser-15 causing cell cycle arrest and senescence to protect cell integrity [228]. Thus, the loss of functional p53 exposes cells to further metabolic stress and oncogenic risk. However, hyperactive p53 can cause premature aging in some tissues [229]. Mice models harboring a p53 mutant associated with modest overactive p53 protein demonstrated an early onset of age-associated diseases including delayed wound healing and reduced hair growth [229,230]. Confirming the role of p53 in regulating longevity, the Arg72 polymorphic variant that predisposes p53 to degradation, and is associated with increased cancer risk, is prevalent in centenarians [231]. This probably implies the activity of other independent tumor suppressor pathways to resist cancer. Aging has been also associated with a decline in antioxidant defense efficiency, which together with higher ROS production significantly contributes to the loss of redox equilibrium [232]. On the other hand, inadequate antioxidant capacity is not necessarily correlated to aging, since the silencing of antioxidant enzymes, such as mitochondrial SOD2 (manganese superoxide dismutase) and GPx-1 (glutathione peroxidase-1), did not affect longevity despite an increased ROS load [225,233]. Catalase, the main hydrogen peroxide detoxifying enzyme, decreases in the dermis and increases in the epidermis of aged and photoaged skin, probably to counteract the augmented ROS load characteristic of stratum corneum [234]. Besides, the function of non-enzymatic components of the antioxidant apparatus is comprised of aging. Lower retinoic acid in old skin partially explains the decreased proliferative potential of aged keratinocytes [235] Among metabolites, Vitamin E, an important skin antioxidant that protects against oxidative damage caused by UV irradiation is strongly reduced in aged skin [235]. Moreover, metabolic studies suggested that aged epidermal keratinocytes shift their energy generation from aerobic respiration in mitochondria to anaerobic glycolysis. This has been attributed to a reduction of endogenously synthesized coenzyme Q10 (CoQ10), a lipid-soluble antioxidant, in the respiratory chain [236].

Chronologic aging of the skin at the cellular level and senescence of cells in vitro are associated with lipoxidizing redox events, mostly related indirectly or directly to ROS [237]. Also, highly reactive lipid oxidation products and their adducts to other macromolecules accumulate in the skin that prematurely aged due to sun exposure [238,239]. The specific lipidomic signature seems to confer resistance to oxidative stress [240,241] indicating a possible implication in the aging process. Longer-lived mammals have more saturated and unsaturated lipids than short-lived species, which is associated with an enrichment in the expression of genes implicated in fatty acids, cholesterol, phospholipids metabolism [242] suggesting that it could be a characteristic acquired under evolutionary selective pressure. Studies regarding lipids in skin aging and diseases mainly focus on the barrier function; however, emerging evidence argues for a role in skin cancers [243]. In vitro studies on primary human dermal fibroblasts and keratinocytes evidenced augmented oxidized phosphatidylcholines (PCs) after irradiation with UVA [244]. Even so, melanin reduces the formation of lipid peroxidation products resulting from UV exposure protecting from photocarcinogenesis [245] suggesting that the final effect mostly depends on the individual phototype. In BCCs, elevated levels of PCs were reported, probably due to the rearrangement of metabolic pathways and increased synthesis of numerous lipids through the conversion of glucose and/or glutamine to cytosolic acetyl-CoA. These processes are thought to support proliferation. However, the role of PCs in skin cancer biology is controversial as oxidized phospholipids demonstrated strong toxicity in melanoma cells and possible therapeutic utility has been proposed [246]. The activity of enzymes that participate in the synthesis and endogenous changes of fatty acids is higher in tumor tissues than in healthy tissues [247]. Transcriptional analysis of BCC demonstrated downregulation of peroxisome proliferator-activated receptor gamma (PPAR-γ) and its transcriptional target genes (*ADIPOQ, FABP4, PLIN1, LPL, ACS, NR4A1, FADS2, HMGCS1*), as well as of additional genes involved in pathways regarding lipid metabolism, unsaturated fatty acid and steroid biosynthesis [248]. Altered metabolic status, characterized by a larger pool of monounsaturated fatty acids, generated by the activity of the Stearoyl-CoA desaturase 1 (SCD1) has been linked to the preservation of melanoma stem cells and to target therapy resistance [247].

### 3.2. Skin Extracellular Matrix and Aging

The structural integrity of human skin is largely dependent on the quality of the dermal ECM that undergoes time-dependent marked changes [249]. From a structural point of view, the effect of intrinsic and extrinsic aging in the skin appears different at clinical and histological levels [250]. Clinically, intrinsically aged skin looks thin and dry with fine lines. In contrast, extrinsically aged skin appears thicker with deep wrinkles. The refinement or thickening of the skin reflects variations in the thickness of the dermal and epidermal compartments. Intrinsic aging also reduces subcutaneous fat with an accompanying loss of cellularity and vascularity [251]. Many histological changes affect structural components of the connective tissue, leading to their increased degradation and the accumulation of nonfunctional matrix due to cross-links in collagen fibers (intrinsic) or to abnormal accumulation of elastin fiber fragments heterogeneous in length (extrinsic). ECM is mostly produced, organized, and maintained by dermal fibroblasts and their senescence drives the entire skin aging process. During aging, dermal fibroblasts progressively enhance the production of metalloproteinases (MMPs) which can degrade matrix components and inhibit collagen synthesis, causing less effective epidermal anchorage, skin relaxation, and decreased interstitial fluid [252,253]. MMP-mediated collagen degradation during aging is antagonized by tissue inhibitors of MMPs (TIMPs) [254]. Thus, TIMPs are considered protective factors against photodamage [255]. In aged cutis, collagen also shows structural alterations characterized by fragmented and unevenly distributed fibrils which contribute to the clinical manifestations of aging [256] Of interest, an inverse correlation between T cell infiltrate and ECM remodeling has been observed partially explaining the poor T cell infiltration rate in the elderly [257]. Other ECM functional components, including fibrillin-rich microfibrils in the papillary dermis, glycosaminoglycans (GAGs), and proteoglycans (PGs), also change during aging. GAGs consist of six types, which encompass chondroitin sulfate (CS), dermatan sulfate (DS), keratan sulfate (KS), heparan sulfate (HS), heparin (HP), hyaluronic acid (HA) and they all seem to have a role in maintaining the hydric skin content due to the presence of negatively charged carboxyl and sulfate groups. HA is especially produced by fibroblasts and its content is significantly reduced in photoaged skin [258]. Photoaging is characterized by the accumulation of disorganized elastic fibers throughout the dermis, a condition known as “solar elastosis” [257]. MMP-2, -3, -9, -12, and -13 could break down elastic fibers. At the same time, in photoaged skin, the regeneration of degraded matrix is also diminished. It has been demonstrated that alterations of ECM occurring with aging and photoaging are permissive per se and not only due to malignant cells’ presence [259]. Actinic keratosis precancerous lesions frequently express MMP1 [260].

Proteins with long lifespans in the dermal matrix and cytoskeleton are particularly susceptible to glycation, leading to reduced elasticity [258]. AGEs react with lysine and arginine residues on collagen molecules causing cross-links between collagen molecules. The levels of AGEs in skin collagen increase linearly with age [261,262] probably due to the irreversibility of this type of modification, the long turnover time, and inefficient collagen repair in aged skin [262,263]. Furthermore, glycated collagen exhibits high resistance to degradation by MMPs (matrix metalloproteinases), resulting in its age-dependent accumulation. Moreover, not only collagen but also elastin, is affected by AGEs, resulting in a reduction of skin elasticity. Glycated elastin fibers exhibit abnormal aggregation and unusual interaction with lysozyme in sun-exposed skin, specifically in cases of solar elastosis, indicating the involvement of glycation in photoaging. Glycosylated vimentin leads to the loss of fibroblast contraction skills and the inability to maintain the cell shape in the elderly [264,265,266]

## 4. Melanoma and Non-Melanoma Skin Cancer Microenvironments: Similar and Peculiar Aspects

In the present section, comparing studies on BCC, SCC, and melanoma we are highlighting some common traits of the cutaneous tumor microenvironment, taking into consideration skin cancer-type differences in terms of disease microenvironment characteristics. The tumor microenvironment is a complex system, which includes various components like the ECM, growth factors, nutrients, blood and lymphatic tumor-serving vessels, and sparse stromal cells that play a crucial role in cancer development and progression [131,132,267]. Dynamically, tumor stroma will allow tumor cells to proliferate, migrate and invade tissues. Non-tumoral stromal cells include tumor-associated macrophages (TAMs) and regulatory T lymphocytes, both having a prominent immunosuppressive action [268]. TAMs are involved in ECM remodeling and neoangiogenesis. On the other side, the immune system additionally plays a critical role in the elimination of pre-malignant and early stages of cancer requiring the cooperation of both the innate and adaptive arms of immunity [269,270,271]. In melanoma, TAMs regulate the expression of hypoxia-inducible factor (HIF)2a in tumor areas where the level of oxygen is low, facilitating tumor cell survival [272]. Further, TAMs release indoleamine 2,3-dioxygenase (IDO) an intracellular enzyme that primes the breakdown of tryptophan in the tumor microenvironment reducing lymphocyte proliferation because of the strong dependence on this amino acid [273]. Changes in melanoma behavior require adaptative modification of neighboring keratinocytes as evidenced by the hyperplastic features of the epidermis surrounding nodular melanoma [274]. Keratinocytes protect melanocytes from malignant transformation releasing ET-1 and α-MSH, which are responsible for the induction of processes of DNA damage repair [275]. Crosstalk between melanocytes and surrounding keratinocytes is essential for skin homeostasis. It is ensured by cell-cell adhesion molecules including connexins and E-cadherin (epithelial cadherin) exposed on the cell surface. A shift from E-cadherin to N-cadherin (neural cadherin) plays a critical role in the regulation of invasion and metastasis, with N-cadherin, allowing melanoma cells to associate preferentially with fibroblasts promoting the invasion of the dermis [276]. E-cadherin loss also releases β-catenin from the membrane increasing its nuclear activity as a co-transcription factor of genes related to cell proliferation [277]. E-cadherin expressing metastatic melanomas had a better outcome when treated with immune checkpoint inhibitor therapy [278]. UVB causes the loss of E-cadherin in melanocytes, leading them to escape from neighboring keratinocytes during melanoma development [279]. Additionally, HGF promotes the shift from E- to N-cadherin contributing to epithelial-mesenchymal transition [279]. UV also lowers cadherin desmoglein1 (Dsg1) expression and consequently melanocyte-keratinocyte interaction, raising the possibility that a Dsg1-deficient niche contributes to pagetoid behavior, such as occurs in early melanoma development [280]. Subtoxic doses of UVA destroy the integrity of gap-junction communication of normal keratinocytes isolating cells from the inhibitory signals of neighboring cells resulting in loss of growth inhibition [281]. In this respect, if the decrease of cell-cell communication by UVA is prolonged, this can contribute to the multi-step process of photocarcinogenesis [281,282]. Interestingly, a recent study reported impaired adhesion in aged keratinocytes due to low E-cadherin expression and reduced calcium-sensing receptor [281] implying a possible inherent inclination of aged skin for tumor cell migration. From the functional point of view, normal keratinocytes in the context of melanoma collaborate with tumor cells facilitating the acquisition of the CAF phenotype by fibroblasts [274]; CAFs are the most abundant stromal cell type in most tumor types including melanoma. CAFs demonstrated a distinct phenotype compared to the “normal” counterpart [130]. Starting from the early tumor stage, due to continuous paracrine stimulation by transformed cells, enclosing stromal fibroblasts are induced to initiate a phenotypic, evolution to transdifferentiate into CAFs. CAFs differ from their normal counterparts by the expression of several markers such as alpha-smooth muscle actin (αSMA), fibroblast-specific protein-1 (FSP-1), fibroblast-activating protein (FAP), platelet-derived growth factor receptors (PDGFR α/β), podoplanin, tenascin-C, desmin, collagen 11-α1 (COL11A1), vimentin, fibronectin, and glutamine-fructose-6-phosphate transaminase 2 (GFPT2), responsible for glycosylation (30734283, 37143134). CAFs have a marked immunomodulating secretome typically activated by IL1β, FGF2, and TGFβ and containing IL6, CXCL8, CXCL1, CCL2, and IFNβ and inflammatory niches have been reported to adapt to and confer drug tolerance to BRAF and MEK inhibitors early during treatment because melanoma cells survive upregulating Bcl2 [283]. BRAF(V600E) melanoma cells expressed higher levels of these cytokines and of MMP-1 than wild-type counterparts and a specific inhibitor of the MAPK pathway, vemurafenib abrogates this condition [284,285]. Further, conditioned medium from the BRAF(V600E) melanoma cells promoted the activation of stromal fibroblasts, an event partially attributable to IL1α/β-mediated modulation of PD-1 ligands in melanoma CAFs [285]. The invasive potential of human melanoma cell lines correlates with the ability to stimulate host stromal fibroblasts, a function that is orchestrated by IL1β expression confirming a key role of this factor in melanoma invasion and metastasis [286]. The interrelationship between high IL6 and IL8 production and melanoma progression has been well documented [287]. IL6 and IL8 also facilitate melanoma cell survival in hypoxia conditions [287]. The relationship between melanoma cells and its stroma is so intricate that the fusion of these cell types forming hybrid clones possibly implicated in tumor recurrence [288]. Normal fibroblasts can counteract the growth and invasion of melanoma by controlling the expression of the MMPs blocking ECM modifications and thus inhibiting the degradation of the basement membrane. On the other hand, senescent fibroblasts have been shown to support melanoma growth and therapy resistance, suggesting that senescence can be a causative cellular mechanism and a risk factor for melanoma [289,290]. Liang and collaborators demonstrated that patients with high expression of cellular senescence-associated genes in associated CAFs and endothelial cells might receive greater benefits from checkpoint inhibitor immunotherapy [291]. On the other hand, melanomas that poorly respond to immune checkpoint blockade had senescence-induced gene loss and amplified senescence suppressors. Melanoma-associated fibroblasts are characterized by upregulated expression of IGF1, VEGFA, FGF2, SDF1, IL6, IL8, CCL2, CXCL12, sFRP2, HAPLN1 and CTGF which support melanoma cell proliferation, survival and drug resistance [292]. MAFs generate an immunosuppressed microenvironment by increasing the expression of inflammatory and immunosuppressive factors, including TGFβ, CCL2, IL6, GM-CSF, MMPs, PGE2, COX2, CXCL5, and PDL1 and PDL2. Furthermore, melanoma CAF can modify the extracellular availability of important immune-modulating metabolites such as lactate, glucose, and arginine, which contribute to immune cell suppression or polarization toward a tumor-promoting phenotype [293].

BCC demonstrated strong stromal dependency [294]. The most predominant genes expressed in BCC are involved in matrix remodeling. The proteases MMP11 and MMP3 are frequently expressed in CAFs of human BCC [295,296,297] whereas MMP10 is expressed in BCCs only in epithelial cancer cells [298,299]. MMP2 and MMP9, having a major role in the digestion of basement membrane collagen IV, are significantly higher in BCCs than in the surrounding normal tissue [300]. MMP1 is strongly present in the stromal and tumor cells of SCC whereas its expression in actinic keratoses correlates to the advanced dysplastic state [301]. Differently, MMP2 and MMP3 are present in SCC stroma but not in actinic keratosis. TIMP-1 and -2 were found in BCC stromal cells [301]. Sustained MMP-1 production requires autocrine EGFR activation [302]. Human SCCs are known to often overexpress epidermal growth factor receptor (EGFR), and the up-regulation of EGFR correlates with increased cell motility and invasion in vitro, often associated with poor prognosis [303,304]. Also the “matricin” DCN was identified as a fibroblast-derived proteoglycan in human BCC potentially exhibiting tumor-promoting properties [305,306]. Fibroblasts involved in BCC express PDGFRβ, collagen XIA, and prolyl-4-hydroxylase (P4H) mRNAs in the tumoral and peritumoral area whereas FAPα is confined in the tumoral lesions [307]. Among inflammatory markers, CXCL12, CCL17, and IL6 are prevalently expressed in the skin surrounding BCC suggesting a role in Treg attraction and immune suppression, although CCL22 and CCL18 are present in both tumoral and peritumoral skin and CCL25 is specifically upmodulated in the BCC lesion [307]. High expression of IL6 in the tumor-near skin, and its absence within BCC might be a result of chronic UV exposure and it is considered an important step contributing to the shift in fibroblast pro-tumorigenic phenotype [307,308]. A comparative analysis of BCC-associated CAFs and fibroblasts derived from normal skin from the same patients, evidenced overexpression of some proteins previously associated with CAF phenotypes and seven BCC-stroma specific proteins (CTSK, cathepsin K, MGP, matrix Gla-protein, CLIP, cartilage intermediate layer protein, DPT dermatopontin, SFRP2, secreted frizzled-related protein 2, PDGFRL, platelet-derived growth factor receptor-like protein, and ANGPTL2, angiopoietin-related protein 2) [309].

SPARC, a matricellular protein implicated in inflammaging could play an important role in skin carcinogenesis. SPARC-null mice are resistant to UV-induced SCC induction, suggesting a tumor-promoting role of SPARC in the skin [310]. In addition, SPARC expression was associated with melanoma progression in nude mice [310,311], and with poor clinical outcome in human small melanoma lesions [312]. In contrast, several studies indicate a clearly opposite tumor suppressive effect of SPARC in several cancer types [313,314]. In xenograft models SPARC inhibited angiogenesis, resulting in reduced tumor growth, which was accompanied by significant alterations in the ECM texture [315].

In normal tissues, the laminins secreted by the epithelial cells form a major component of the basal lamina which functions to anchor the epithelium to the underlying stroma [316]. Laminin 332, previously known as laminin 5, is highly expressed in several types of SCC and other epithelial tumors [317]. Laminin receptor expression, such as integrins, have also been shown to play an important role in SCC progression by regulating processes such as adhesion and migration [318,319]. Chemerin, released by senescent dermal fibroblasts, exerts a strong chemotactic activity in cSCC cells and thus contributes to enhanced SCC cell migration via activation of the MAPK pathway. These findings corroborate the observation that the putative Chemerin receptor, CCRL2, is remarkably upregulated in SCC tumor cells, which mediates the response to Chemerin. CCRL2 upregulation confers a migratory advantage for the epidermal SCC tumor cells, particularly at the invasive front of the epidermis-dermis junction, facilitating their migration towards the dermis, where a gradient of its high-affinity ligand, Chemerin, is established by senescent fibroblasts [320].

The capacity of fibroblasts to sustain cancer largely depends on the ability to synthesize and secrete a large variety of pro-growing peptides that facilitate tumor growth [321,322,323,324]. Specifically, augmented expression of b-FGF, HGF, SCF, and VEGF by melanoma CAF respect normal fibroblasts confirming the pro-mitogenic attitude of melanoma-associated fibroblasts [130]. However, the relevance of growth factors in SASP is still debated [12]. Since up- and down-modulation of growth factors seem to be the main difference between stress-induced premature senescence and physiological aging of the skin it is not senseless that only extrinsic aging or neoplasm onset might force fibroblast to growth factors hypersecretion.

## 5. Discussion

Cutaneous aging is a complex progressive process resulting from the combination of intrinsic and extrinsic factors. Aging of the skin affects some important physiological functions necessary for tissue homeostasis. Skin-aged phenotypes include easily visible signs such as fine lines, loss of volume, deep wrinkles and non-homogeneous pigmentation. However, one of the most dangerous consequences of aged-associated loss of tissue equilibrium is the predisposition to develop infection, autoimmune and hyperproliferative diseases. Evidence has also emerged that slowing down aging can in turn delay cancer occurrence, improve cancer prognosis, and increase the overall quality of cancer patients’ lives. Thus, the importance of skin aging prevention strategies overcomes the simple aesthetic interest and it has been proposed as an innovative approach for NMSC [135]. Hence, from the therapeutical point of view, extrinsic mechanisms of aging appear to be more targetable since genetically determined and time-related changes in the skin are difficult to manipulate. Consistently, photoprotection is the first-line strategy to counteract premature cutaneous aging. Topic and systemic antioxidant supplementation are valid options to slow ECM rearrangement and dermal atrophy. The advantage of anti-aging agents consists in the possibility to target both normal post-mitotic cells (i.e., aged cells, senescent matrix, immunosenescence) as well as neoplastic cells, at least in theory without resistance hindrance. Several studies documented that the application of senolytics (especially natural products) as adjuvants to other treatments during melanoma therapy has positive implications for melanoma prognosis. Moreover, other studies have shown that applying senostatics (resveratrol) can reduce the amount of SASP factors released by senescent fibroblasts inhibiting the stimulatory effect of these cells on melanoma cell proliferation and invasion [136]. However, further information regarding the role of senescence in the context of cancer, particularly in nevi and melanoma onset, is necessary to proceed from the preclinical setting to the clinical one.

## 6. Conclusions

Given the relevance of aging in skin cancer, understanding the mechanisms that link skin aging and malignant transformation is crucial for developing preventive therapies. Senescence, together with apoptosis, is a powerful barrier against neoplastic progression, however senescent phenotype in the stroma surrounding neoplasm offers several supporting factors confounding the immune system and compromising the physiological cell cross-talk. Mostly, pro-tumorigenic aged-associated features are related to extrinsic insults, particularly UV suggesting that lifestyle might be effective in skin cancer prevention.

## Figures and Tables

**Figure 1 ijms-24-14043-f001:**
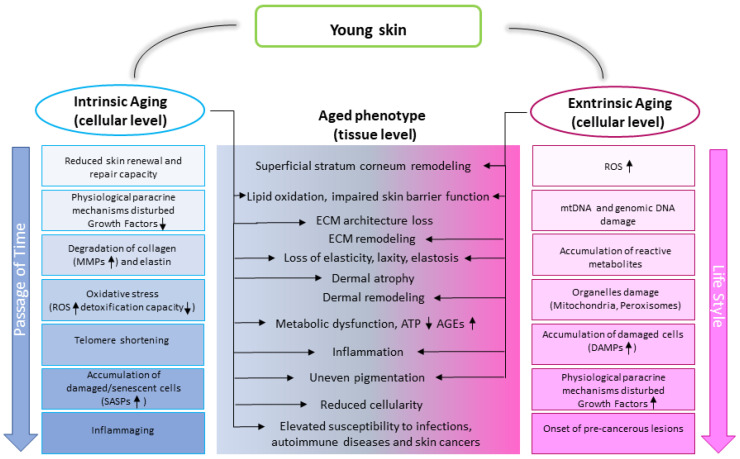
Schematic representation of intrinsic (internal) and extrinsic (external) factors and of associated skin-aged phenotype.

**Figure 2 ijms-24-14043-f002:**
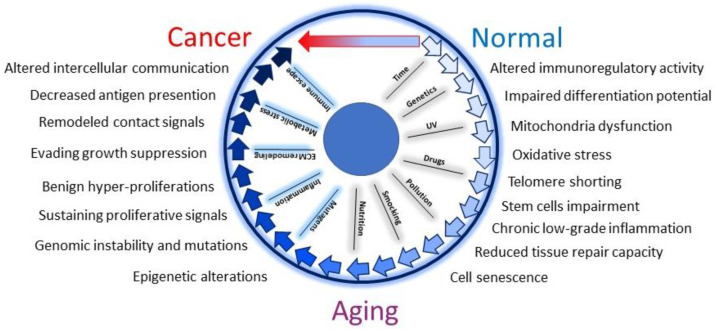
Schematic illustration of various mechanisms involved skin aging, in cancer onset and cancer-associated fibroblast (CAF) activation.

**Figure 3 ijms-24-14043-f003:**
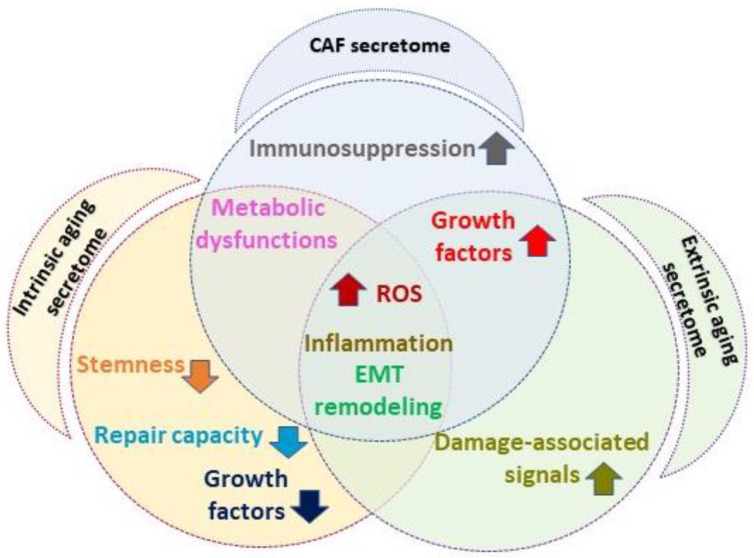
Representation of relevant overlapping and divergent characteristic of intrinsically aged skin, extrinsically aged skin and of CAFs.

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
