# Peer review of "Skin Cancer Microenvironment: What We Can Learn from Skin Aging?"

_ijms, 2023, doi:10.3390/ijms241814043_

Round 1

Reviewer 1 Report

The authors performed a review of the current literature with the aim of discussing changes that occur during aging that can prime neoplasm or increase the aggressiveness of melanoma and non-melanoma skin cancer. The manuscript is interesting and well-written. Moreover, the theme of the review is very interesting, leading to the development of several interesting points of reflection which may affect daily clinical practice.

I have only few suggestions.

My comments:

- Introduction: nothing to add, this section is complete offering readers a clear overview of the treatment

- Paragraph 2 is repeated twice, please renumber the sections

- Material and Methods: this section should be added, in order to explain how you performed literature review.

- Discussion section: a discussion section briefly summarizing your results and reporting an expert opinion on possible clinical applications of your study should be reported. Moreover, I suggest briefly report currently treatment options  (doi:10.1007/s12325-022-02044-1 and doi:10.3390/ijms23126388) in order to evaluate possible therapeutic applications of your results.

- strengths and limitations of the study should be discussed

Author Response

Editorial Office

International Journal Molecular Science                                      Rome, 29th August 2023

Revised manuscript IJMS-2569434

Dear Editor,

please find enclosed the revised version of the invited Review article entitled “Skin cancer microenvironment: what we can learn from skin aging by Andrea D’Arino et al. Firstly, we would like to thank the reviewers for the positive and helpful comments about our paper. We have carefully considered all comments and we have revised the manuscript accordingly to the requests and answered the comments point-by-point as follows:

Reviewers' comments:

Reviewer 1

The authors performed a review of the current literature with the aim of discussing changes that occur during aging that can prime neoplasm or increase the aggressiveness of melanoma and non-melanoma skin cancer. The manuscript is interesting and well-written. Moreover, the theme of the review is very interesting, leading to the development of several interesting points of reflection which may affect daily clinical practice.

I have only few suggestions.

My comments:

- Introduction: nothing to add, this section is complete offering readers a clear overview of the treatment

- Paragraph 2 is repeated twice, please renumber the sections

The numbering of sections has been corrected.

- Material and Methods: this section should be added, in order to explain how you performed literature review.

As requested a Material and Methods section has been added to explain that the Review has been designed as a narrative review (or simply a literature review) without any systematic methods.

- Discussion section: a discussion section briefly summarizing your results and reporting an expert opinion on possible clinical applications of your study should be reported. Moreover, I suggest briefly report currently treatment options (doi:10.1007/s12325-022-02044-1 and doi:10.3390/ijms23126388) in order to evaluate possible therapeutic applications of your results.

As requested, we added a short discussion including possible therapeutic intervention in terms of chemoprevention.

- strengths and limitations of the study should be discussed.

This aspect has been included in the Material

Reviewer 2 Report

Dear Authors, 

Abstract: contains information summarizing the whole article, interesting.

Introduction: contains the right information, however, such a large number of words should be divided into paragraphs 1.1. , 1.2 etc., well-chosen literature

In the part of the proper review, the mechanisms are described, but there are no graphics related to this subject. I believe that it is important to present at least a few mechanisms, both chemical reactions responsible for aging and processes involving organelles at the cellular level.

No chemical reactions at the cellular  level, no description of the impact of energy balance and cellular organelles (pathway of the process)

Thank you.

I suggest re-editing Fig 2 to make the editing more biological than mathematical.

Author Response

Editorial Office

International Journal Molecular Science                                      Rome, 29th August 2023

Revised manuscript IJMS-2569434

Dear Editor,

please find enclosed the revised version of the invited Review article entitled “Skin cancer microenvironment: what we can learn from skin aging by Andrea D’Arino et al. Firstly, we would like to thank the reviewers for the positive and helpful comments about our paper. We have carefully considered all comments and we have revised the manuscript accordingly to the requests and answered the comments point-by-point as follows:

Reviewers' comments:

Reviewer 2

Dear Authors, 

Abstract: contains information summarizing the whole article, interesting.

Introduction: contains the right information, however, such a large number of words should be divided into paragraphs 1.1. , 1.2 etc., well-chosen literature

As suggested, in the revised manuscript the the one-paragraph introduction has been organized into three different paragraph.

In the part of the proper review, the mechanisms are described, but there are no graphics related to this subject. I believe that it is important to present at least a few mechanisms, both chemical reactions responsible for aging and processes involving organelles at the cellular level.

As requested, a graphical representation of major mechanisms involved in skin aging at cellular and tissue level has been prepared.

No chemical reactions at the cellular level, no description of the impact of energy balance and cellular organelles (pathway of the process)

We did not include details of chemical reaction and individual pathways since the aim of the review to offer a more general overview of the topic.

Thank you.

I suggest re-editing Fig 2 to make the editing more biological than mathematical.

We apologize, the request is not fully clear since the graphic shown senescence-associated feature of intrinsically or extrinsically aged cells, features specific of CAF phenotype and aspects that are shared by both cell type (merged area). All the legend indicate biological process.

We hope that this revised version is now suitable for publication in IJMS. Thank you for your time and effort on our behalf.

Best regards,

Barbara Bellei

Round 2

Reviewer 1 Report

The authors performed a review of the current literature with the aim of discussing changes that occur during aging that can prime neoplasm or increase the aggressiveness of melanoma and non-melanoma skin cancer. The manuscript is interesting and well-written. Moreover, the theme of the review is very interesting, leading to the development of several interesting points of reflection which may affect daily clinical practice. Finally, all the reviewers' comments have been addressed, leading to an improvement of the manuscript's quality.

The article is now suitable for publication.

Reviewer 2 Report

Thank you